# Adaptive Motion Artifact Reduction in Wearable ECG Measurements Using Impedance Pneumography Signal

**DOI:** 10.3390/s22155493

**Published:** 2022-07-23

**Authors:** Xiang An, Yanzhong Liu, Yixin Zhao, Sichao Lu, George K. Stylios, Qiang Liu

**Affiliations:** 1Academy of Artificial Intelligence, Beijing Institute of Petrochemical Technology, Beijing 102617, China; anxiang@bipt.edu.cn (X.A.); 2020520121@bipt.edu.cn (Y.L.); zhaoyixin@bipt.edu.cn (Y.Z.); 0020210007@bipt.edu.cn (S.L.); liuq@bipt.edu.cn (Q.L.); 2Beijing Academy of Safety Engineering and Technology, Beijing 102617, China; 3Research Institute for Flexible Materials, School of Textiles and Design, Heriot-Watt University, Galashiels TD1 3HF, UK

**Keywords:** electrocardiogram, impedance pneumography, motion artifact reduction, adaptive filtering

## Abstract

Noise is a common problem in wearable electrocardiogram (ECG) monitoring systems because the presence of noise can corrupt the ECG waveform causing inaccurate signal interpretation. By comparison with electromagnetic interference and its minimization, the reduction of motion artifact is more difficult and challenging because its time-frequency characteristics are unpredictable. Based on the characteristics of motion artifacts, this work uses adaptive filtering, a specially designed ECG device, and an Impedance Pneumography (IP) data acquisition system to combat motion artifacts. The newly designed ECG-IP acquisition system maximizes signal correlation by measuring both ECG and IP signals simultaneously using the same pair of electrodes. Signal comparison investigations between ECG and IP signals under five different body motions were carried out, and the Pearson Correlation Coefficient |r| was higher than 0.6 in all cases, indicating a good correlation. To optimize the performance of adaptive motion artifact reduction, the IP signal was filtered to a 5 Hz low-pass filter and then fed into a Recursive Least Squares (RLS) adaptive filter as a reference input signal. The performance of the proposed motion artifact reduction method was evaluated subjectively and objectively, and the results proved that the method could suppress the motion artifacts and achieve minimal distortion to the denoised ECG signal.

## 1. Introduction

In recent years, there has been an increasing interest in wearable electrocardiogram (ECG) monitoring systems. This interest comes from two aspects: societal needs and technology drive. Wearable ECG monitoring systems provide real-time vital sign monitoring, early disease detection, and telemedicine interventions, providing a good solution to the medical challenges posed not only by rapid population aging and healthcare cost inflation, but also by wellbeing requirements of an increasingly more health-conscious society. There is a huge demand for the latter, because as people are becoming more aware of their wellbeing and doing more exercise, these devices help them conduct their own, personal health monitoring independently. This has become possible with the miniaturization of electronics, improvements in the performance of low-power microprocessors, the ability of electronic textiles, and the development of artificial intelligence, rendering wearable healthcare monitoring systems to become increasingly portable, accurate, and powerful.

The ECG signal reflects the electrical activity of the heart, which is a weak physiological signal and hence very susceptible to various kinds of noise [1]. However, unlike the resting ECG measurements, wearable ECG measurements often face electromagnetic interference caused by the complex electromagnetic field environment in daily life, as well as motion artifact interference and EMG interference caused by various movements of the human body. The presence of noise in ECG signals may corrupt the ECG waveform causing inaccurate ECG interpretation, which in turn can lead to misdiagnosis of heart disease, inappropriate treatment decisions, and trigger false alarms. Therefore, to prevent inaccurate ECG waveforms and misdiagnosis of heart disease, reducing the noise in the ECG signal is a very important requirement before ECG signal analysis, especially for wearable ECG monitoring.

Motion artifacts have long been a problem in ECG measurement, the reduction of motion artifacts is difficult and challenging not only because the spectrum of motion artifacts is considerably overlapped with the important spectral components of the ECG signal, but also because the presence of the motion artifacts is inevitable and unpredictable in the wearable ECG measurement data. Motion artifacts are transient (but not step) baseline wander resulting from movement of the wearer, or the electrode [2]. The most widely accepted explanation of motion artifacts is from Tam and Webster [3], who found that the change in skin potential at the skin-electrolyte interface is the main source of motion artifacts. According to Edelberg’s skin model [3], the skin potential is determined by the potential across the sweat duct membrane, the total resistance in the sweat duct, the potential across the epidermis barrier membrane, and the total series resistance of the epidermis. Changes in any of the four variables will cause changes in skin potential.

The skin potential is usually negative, and the magnitude of the skin potential varies depending on the subject, skin site, and the condition of the skin. Floyd and Keele [4] found that increasing gel concentrations of NaCl or KCl increase the magnitude of the skin potential, and that the deformation of skin is the main cause of motion artifacts. Tam and Webster [3] found that the skin potential changes with pressure on the skin and when the skin stretches, the skin potential decreases. “If the thickness of the skin layer is changed by stretching or pressing down on the skin, the skin potential can vary by 5–10 mV compared to the 1 or 2 mV ECG signal” [3], which causes rapid changes in the ECG baseline. Consequently both the wearer’s movement and the respiration can cause deformation of the skin and hence introduce motion artifacts to the ECG signals.

To make wearable ECG monitoring systems practically applicable, motion artifact reduction is one of the most challenging problems that need to be overcome. The performance of different motion artifact reduction methods has been investigated in many studies [5,6,7,8,9,10,11,12,13,14,15,16,17,18], and adaptive filters have been shown to be useful in motion artifact reduction in some of those studies [19,20,21,22,23,24]. Tong et al. [25] used an anisotropic magnetoresistive (AMR) sensor and an accelerometer as the source of reference input, and their results indicate that adaptive filtering can reduce the amount of motion artifact present in the ECG, and the accelerometer-based sensor outperforms the AMR sensor. There are also some other reports that have used an accelerometer as a source of noise reference for adaptive motion artifact reduction; Raya and Sison [26] attached an accelerometer to the wearer’s back, Liu [24] placed an accelerometer next to the left leg electrode, and Lee et al. [27] placed an accelerometer between two electrodes. However, in a study of a wearable monitoring system based on textile electrodes, Zhang et al. [20] reported that motion artifacts were not effectively reduced by adaptive filtering with the accelerometer outputs as reference signals. The skin-electrode impedance signal was also reported to be a useful source as the reference signal for adaptive filtering [28,29,30]. Although the results from most literature prove that the adaptive filter can be used to reduce motion artifacts, there is no conclusion to determine which reference input signal can most effectively reduce motion artifacts caused by different motions [31], and some findings are contradictory, whilst others inconclusive.

In this article, we aimed to use an adaptive filtering technique to solve the problem of motion artifact in wearable ECGs. We have studied the motion artifact reduction performance of adaptive filters that use an impedance pneumography (IP) signal as a reference input signal. To ensure that the IP signal correlated with the motion artifact in ECG signals, we designed hardware that records ECG and IP simultaneously. The correlation between the IP and the motion artifact in the ECG signal was studied in order to test whether the IP signal is suitable as the reference input for adaptive filtering. Finally, we evaluated the performance of the proposed motion artifact reduction method with noisy ECG signals that were recorded while the subjects performed five typical upper body motions. 

## 2. Methodology

### 2.1. Adaptive Filtering

The adaptive filtering method attracted our attention because it can adaptively track the signal under non-stationary conditions and can adjust its impulse response to filter out the noise in the input with little or no prior knowledge of the signal and noise characteristics [32]. The adaptive filter works generally for the adaptation of signal-changing environments, the spectral overlap between noise and signal, and unknown or time-varying noise [33]. It has the capability of adaptively tracking the signal under non-stationary conditions and can be used for different purposes, such as system identification, prediction, and noise cancellation.

Figure 1 demonstrates the working principle of the adaptive filter in noise cancellation. In adaptive noise cancellation systems, the objective is to produce a system output *X’*(*n*) that is the best fit in the least squares sense to the signal *X*(*n*) by feeding the system output back to the adaptive filter and adjusting the filter through an adaptive algorithm to minimize total system output power [32]. The adaptive filter contains a digital filter with adjustable coefficients and the adaptive algorithm to modify the values of coefficients for filtering each sample. The coefficients of the digital filter are continuously changed according to the chosen adaptive algorithm so as to minimize the mean squared value of the error signal *e*(*n*). The reference signal *S*(*n*) is fed into a digital filter to produce an output *N’*(*n*)*,* which is as close as possible to the replica of the noise *N*(*n*). Subsequently, this filtered signal output *y*(*n*) is subtracted from the primary input *d*(*n*) to obtain the estimated desired signal *X’*(*n*) [5]. As can be seen from Figure 1, the performance of an adaptive filter in noise reduction is mainly determined by the adaptive algorithm and the reference input. 

Least mean squares (LMS), Normalised Least Mean Squares (NLMS), and Recursive Least Squares (RLS) are the three most commonly used algorithms in adaptive filtering. In our preliminary trials, we found the three adaptive algorithms LMS, NLMS, and RLS are all capable of removing motion artifacts from the noisy ECG signal, but their noise reduction performance varies with the selection of the step size μ, or the forgetting factor λ. In our proposed adaptive motion artifact reduction method, we adopted the RLS algorithm because it provides a faster convergence speed. Furthermore, the forgetting factor of the RLS algorithm was reduced from 1 to 0.99 to improve the tracking ability of the adaptive filter. However, the reduction of forgetting factor will sacrifice the stability of the algorithm and thus cause a high-frequency noise in the denoised ECG, as demonstrated in Figure 2a, so we added a 5 Hz low-pass filter to the estimated noise *N’*(*n*) to reduce the unwanted high-frequency components, the results of the denoised ECG using an RLS adaptive filter and a 5 Hz low-pass filter are shown in Figure 2b. The cut-off frequency of the low-pass filter was set to 5 Hz because the main spectral range of the motion artifacts we wanted to suppress is 0–5 Hz [34]. 

As for the reference input signal, according to the principles of adaptive filtering, a suitable reference input signal should correlate in some way with the noise to be removed but uncorrelated with the desired signal [32]. However, in some practical cases, the available reference input signal may contain uncorrelated noise components in addition to the correlated noise components or contain low-level of desired signal components. When there is uncorrelated noise presented in the primary and reference input signals, the performance of the adaptive filter in noise cancellation will be degraded. In other words, a reference input signal that is highly correlated with the noise in the primary input is preferred. Moreover, when there are signal components presented in the reference input, some signal distortion will occur and hence the performance of the adaptive filter in noise cancellation will be degraded. The more signal components present in the reference input, the higher the signal distortion in the output. Therefore, in order to have a small signal distortion in the output, a reference input that contains small signal components is desired. In the application of adaptive motion artifact reduction in wearable ECGs, the reference input signal should come from a relatively independent source rather than noisy ECG signals to prevent the suppression and distortion of the ECG waveform. 

In summary, a desired reference input signal for motion artifact reduction should have a good correlation with motion artifacts and should not be correlated with the ECG signal itself. Therefore, in our proposed method, we used specially designed hardware to simultaneously record both the ECG signal and the reference input signal in order to maximize the correlation between the primary input and the reference input. Furthermore, we applied a 5 Hz low-pass filter to eliminate uncorrelated noise and unwanted ECG components from the reference input signal. The proposed adaptive motion artifact reduction method is shown in Figure 3.

### 2.2. Impedance Pneumography

The Impedance Pneumography (IP) signal is a measure of the electrical impedance change of the subject’s thorax caused by respiration, as any increase in the volume of air respiration during each breathing cycle the electrical conductivity of the person’s thorax is reduced. In fact, if we consider the IP signal generated by the change in impedance as noise, it reflects the respiratory motion artifact as well as the motion artifact caused by skin deformation. Thus, from the origin of IP signals, there is a certain correlation between IP signals and the motion artifacts in ECG signals, which makes the IP signal a good option when looking for a reference input signal to adaptive reduce motion artifacts. 

In order to suppress motion artifacts from ECG signals, we designed a wearable ECG-IP acquisition system consisting of a microcontroller MSP430F5438A (Texas Instruments, Dallas, TX, USA), an analogue front-end (AFE), a Bluetooth module RN41 (Microchip, Chandler, AZ, USA), and three textile electrodes. The AFE circuit board contains an analogue pre-amplifier ADS1292R (Texas Instruments, Dallas, TX, USA) and its peripheral circuitry to simultaneously record ECG and IP signals. The ADS1292R used in the AFE is of low power, and it is characterized by a high common-mode rejection ratio (−120 dB), 2-channels, and a 24-bit delta-sigma analog-to-digital converter (ADC) with integrated programmable gain amplifiers (PGAs). It is able to sample the ECG and IP signals, convert them from analog to digital and send data via a Serial Port Interface. With the integration of the respiration impedance measurement module, the ADS1292R measures the sum of the skin-electrode impedance and thoracic impedance. The respiration module in the ADS1292R contains a modulation block and a demodulation block. The modulation block produces a 2.42 V, 32 kHz square wave and injects it into the body as a carrier that is amplitude-modulated by a low-frequency signal generated as a result of the breathing action. The generated 32 kHz square wave is fed into the electrodes through the RESP_MODP and RESP_MODN pins, and the amplitude-modulated signal is received by the IN1P and IN1N pins of the ADS1292R. The measurement of the respiration impedance uses the same pair of electrodes for ECG measurement, so the measured IP signal has some correlation with the motion artifacts in the ECG signal. Figure 4 shows the designed analog front-end (AFE) circuit board. The configuration of the peripheral circuit of ADS1292R is referred to the recommendation of the technical datasheet by Texas Instruments [35]. 

Figure 5 shows examples of the recorded IP signal by the designed ECG -IP acquisition system.

As can be seen in Figure 5a, the measured IP signal contains not only the respiratory impedance signal but also the cardiac-related artifacts, as the transthoracic electrical impedance varies with both respiration and cardiac function [36]. The respiratory impedance signal is caused by the changes in lung volume during respiration because an increase in the amount of air in the lung reduces electrical conductivity. The cardiac-related artifacts [37] are mainly generated by changes in the blood volume during every cardiac cycle and have some correlation with ECG signals. In addition to the cardiac-related artifacts, the IP signal is also susceptible to body movement, as can be seen in Figure 5b. Changes in chest shape and/or in the skin-electrode impedance caused by body movement can produce large motion artifacts in the IP signals. In some cases, the motion artifacts are higher than the respiratory-related signals.

The aim of this work was to reduce the motion artifacts in the ECG signal by adaptive filtering, so the feasibility of using an IP signal as a reference input signal to adaptively reduce motion artifacts from the ECG signal was discussed. Since the ideal reference input for an adaptive filter should be correlated with the motion artifacts to be canceled but uncorrelated with the ECG signal, the motion artifacts in the IP signal should be preserved, but the cardiac-related signal should be eliminated. The frequency of cardiac-related artifacts in the IP signals is usually related to the QRS complex of ECG waves, which has a fundamental frequency of about 10 Hz [38]. As for the frequency of motion artifacts in IP signals, it is usually determined by the frequency of body motion, and the major frequency range of daily activities is 0.3–3.5 Hz [34]. Therefore, in our proposed adaptive motion artifact reduction method, we applied a 5 Hz low-pass FIR filter to the IP signal to suppress its cardiac-related artifacts before feeding it to the adaptive filter, as shown in Figure 3. 

## 3. Experiments and Results

### 3.1. Experimental Signal Measurement

In the experiments, three textile electrodes were used to measure wearable ECG and IP signals, two of which were connected to the ELA and ERA pin of the AFE circuit board for 1-lead ECG recording and IP recording, and the remaining one textile electrode was connected as a ground electrode to the ERL pin of the AFE circuit board for common mode noise rejection. The fabrication of textile electrodes and their performance for ECG measurement has been discussed in another of our articles [39]. The three textile electrodes were sewn onto a knitted heavy stretch elastic band, as seen in Figure 6. During the measurement, the electrode chest band was placed on the horizontal line of the 6th intercostal space, where the electrodes connected to ELA and ERA were placed on the left and right midclavicular line, respectively, and the ground electrode was placed on the subject’s back. 

In order to evaluate the performance of our proposed adaptive motion artifacts reduction method with the actually recorded noisy ECG signals, we simulated five common motions of daily life, and used the designed ECG-IP acquisition system to record the raw ECG and IP signals as disturbed by the different types of motion, as shown in Figure 7. A detailed description of these five types of motion is listed in Table 1. Ten female subjects and ten male healthy subjects between the ages of 21–26 participated in the experiment, each subject repeated each of the five types of motions ten times. 

As shown in Table 1, Motion type I is to simulate the interference of external force on the electrodes. Motion types II–IV are to simulate three basic types of upper body movement. Motion type V is to simulate the respiration movement in a static state. In daily activities, various actions of the human body are mainly accomplished through the cooperation of muscles and joints of different body segments (such as upper limbs, upper body, and legs). During ECG measurement, because ECG electrodes are placed on the subject’s chest, only upper-body motions are more likely to produce motion artifacts. Therefore, in this study, three basic upper-body motions were selected. 

Figure 8 shows the typical ECG and IP signals recorded in the experiment. It can be seen from Figure 8a that pressing the electrode causes severe baseline drift of the ECG signal, and the IP signal was also severely disturbed. The result indicates that the ECG and IP signals are very sensitive to external interference on the electrodes. As can be seen from Figure 8b–d, the upper body movement causes a certain amount of baseline drift in the ECG and IP signals. Figure 8e shows the typical signals measured when the subject was standing still, in which the respiratory motion artifact can be easily observed.

### 3.2. Signal Correlations

According to the principles of adaptive filtering, the performance of an adaptive filter in reducing motion artifacts is highly dependent on the characteristics of the reference signal. The higher the correlation between the reference signal and the motion artifacts, the better the performance in reducing motion artifacts. Therefore, the Pearson correlation coefficient was used as an indicator to determine whether the IP signal can be used as a reference input signal for the motion artifacts reduction.

The absolute value of the Pearson correlaion coefficient |r| is described as:(1)$$\left|r\right|=\left|\frac{\mathrm{COV}\;\left(x,y\right)}{\sigma_{x}\sigma_{y}}\right|=\left|\frac{E\left(\left(x-\mu_{x}\right)\left(y-\mu_{y}\right)\right)}{\sigma_{x}\sigma_{y}}\right|$$
where x is the 5 Hz low-pass filtered noisy ECG signal, y is the 5 Hz low-pass filtered IP signal; COV (x,y) is the covariance and E is the expectation; $\mu_{x}$, $\sigma_{x}$ are the mean and standard deviation of x; $\mu_{y}$, $\sigma_{y}$ are the mean and standard deviation of y. Note that in the equation, x should be a motion artifact signal, but in practice, the motion artifact signal is not independent but is included in the noisy ECG signal, so a 5 Hz low-pass filtered noisy ECG signal is used instead of the motion artifact signal to calculate the Pearson correlation coefficient. When |r|>0.7, it indicates a strong correlation; when 0.5<|r|<0.7, it indicates a moderate correlation; when 0<|r|<0.5, it indicates a weak correlation.

Figure 9 shows the calculated mean and standard deviation of the absolute correlation coefficient (|r|) under five body motions. The mean and standard deviation were calculated from the 200 measurements of each motion. As can be seen from Figure 9, the IP signal has a good correlation (|r| > 0.6) with noisy ECG signals in most body motions. The results indicate that the low-pass filtered IP signal has a good correlation with the corresponding motion artifacts, so it can be used as a reference input signal for adaptive motion artifact reduction. 

### 3.3. Evaluation Parameters

Parameters such as the improvement of signal-to-noise ratio (SNR), correlation coefficient (c), mean-squared error (MSE), and R-square were often used to evaluate the noise reduction performance of different filters. However, the calculation of these parameters requires prior knowledge of the “clean” signal, which is the main reason that the simulated noisy ECGs were widely used in other relevant studies. But in our work, the actual recorded noisy ECGs were used instead of the simulated noisy ECGs, so there is no independent noise-free “clean” ECG signal that can be used to calculate parameters such as SNR. 

Therefore, in this study, the concept “goodness-of-fit” [40] was used to evaluate the performance of the proposed adaptive motion artifact reduction method. According to the principles of adaptive filtering, the denoised ECG signal is the noisy ECG signal minus the estimated noise. When the estimated noise is close enough to the actual noise, the adaptive filter has good noise reduction performance. Thus, the goodness-of-fit between the estimated noise and the actual noise is an alternative measure to evaluate the noise reduction performance of adaptive filtering. To measure goodness-of-fit between the estimated noise and the actual noise, the estimated noise signal and the actual noise signal are essential. The estimated noise signal is the filtered signal output y(n), as described in Figure 3. The actual noise is obtained by low-pass filtering of the noisy ECG signal using an IIR filter with a cut-off frequency of 5 Hz. The 5 Hz low-pass filtered noisy ECG signal contains most motion artifacts but excludes most of the QRS components of the ECG signal. Two parameters were used to measure the “goodness of fit”, as denoted in the equation: (2)$$m\_\mathrm{MSE}=\frac{1}{n}\sum_{i=1}^{n}{\left(d_{\mathrm{lowpass}\left(i\right)}-y\left(i\right)\right)}^{2}$$
(3)$$m\_R^{2}=1-\frac{\sum_{i=1}^{n}{\left(d_{\mathrm{lowpass}\left(i\right)}-y\left(i\right)\right)}^{2}}{\sum_{i=1}^{n}{\left(d_{\mathrm{lowpass}\left(i\right)}-\;{\bar{d}}_{\mathrm{lowpass}}\right)}^{2}}$$
where $d_{\mathrm{lowpass}\left(i\right)}$ is the 5 Hz low-pass filtered noisy ECG signal, y(i) is the estimated noise. The m_MSE value closer to 0 indicates a better fit, and the m_R2 value closer to 1 indicates a better fit.

The performance of the adaptive motion artifact reduction method was also compared with the 0.5 Hz IIR high-pass filter commonly used in practice to verify the efficiency of the proposed method. For this comparison, the standard deviation of the original ECG signal and the standard deviation of the denoised ECG signal were used to measure the reduction of motion artifacts. In statistics, the standard deviation is a measure of the amount of variation in a set of values. Because the motion artifacts are reflected as ECG baseline drift, a low standard deviation indicates less baseline drift (fewer motion artifacts), and a high standard deviation indicates more baseline drift (more motion artifacts). Therefore, the performance of the adaptive motion artifact reduction method and the IIR filter can be evaluated by comparing the STD values of the original noisy ECG and the denoised ECG. 

The standard deviation (STD) is defined as:(4)$$\mathrm{STD}=\sqrt{\frac{1}{N-1}\sum_{i=1}^{N}{\left|x_{i}-\;\bar{x}\right|}^{2}}$$
where $x_{i}$ is the value of the sample, $\bar{x}$ is the mean value of samples, N is the number of the samples. 

### 3.4. Results and Discussion

Figure 10 shows the examples of the denoised ECG signal (red) compared to the original noisy ECG (blue). As can be seen from the graphs, the proposed adaptive motion artifact reduction method has good performance in reducing motion artifacts and thus preventing ECG waveform distortion. For objective evaluation of the performance of the proposed method, parameters m_MSE and m_R^2^ were calculated. The m_MSE value closer to 0 and the m_R^2^ value closer to 1 indicate that the performance of motion artifact reduction is better. Results are shown in Figure 11 and Figure 12.

As can be seen in Figure 11 and Figure 12, the m_MSE is smaller than 0.15, and the m_R^2^ is larger than 0.7. These results indicate that the proposed adaptive motion artifact reduction method has good performance in reducing motion artifacts. Figure 13 shows the results of the standard deviation (STD) of the original noisy ECG signal, IIR filtered ECG signal and the adaptive filtered ECG signal. It can be seen from the results that the STD of the noisy ECG signal is significantly reduced by both methods. The proposed method has a lower STD than the IIR filter, which proves that the proposed method can effectively reduce the motion artifact in ECG signals. Furthermore, the adaptive motion artifact reduction method has a distinct advantage over the high-pass IIR filter in reducing motion artifacts from ECG signals because it can adaptively track the baseline drift while preserving the ECG waveform. 

## 4. Conclusions

Driven by social needs and technology, the demand for wearable healthcare monitoring devices is growing and is expected to accelerate in the future. Wearable ECG devices have attracted a lot of attention because it makes long-term ambulatory ECG monitoring possible, which helps to identify arrhythmias that are not detected by resting ECG or Holter ECG systems. However, due to the mobility of the human body and the various types of body movement in daily life, motion artifacts are a common and persistent problem for effective wearable ECG monitoring. The occurrence of motion artifacts can corrupt the ECG waveform causing inaccurate signal interpretation, so reducing their presence is necessary.

In this paper, an adaptive motion artifact reduction method based on an improved RLS adaptive filter is proposed to solve the motion artifact problem in wearable ECGs. In this method, the IP signal was used as the reference input signal of the improved RLS adaptive filter. In order to ensure correlation between the reference input signal IP and the motion artifacts in the ECG signal, we have specially designed the ECG-IP synchronous acquisition system discussed, to record both the ECG and the IP signal through the same pair of electrodes. To further improve the signal correlation, we applied a 5 Hz low-pass filter on the IP signal to eliminate the cardiac-related signal in the IP signal while preserving motion artifacts. According to the signal correlation study, we found that the processed IP signal has a good correlation (|r| > 0.6) with the motion artifacts, which proves that the IP signal is suitable as the reference input signal of the adaptive filter for motion artifact reduction. 

Supported by the specially designed ECG-IP acquisition system, the performance of the proposed adaptive motion artifact reduction method was investigated by using real recorded noisy ECG signals. By analyzing the denoised ECG signals, it was found that most motion artifacts were removed and little distortion can be noticed from the denoised ECG waveform. By calculating the evaluation parameters, it was found that the mean squared error (m_MSE) is smaller than 0.15 and the R-square (m_R^2^) is larger than 0.7, which indicates that the estimated noise by the proposed method has a good fitness with the actual motion artifact. Finally, the results prove that the adaptive motion artifact method can suppress most of the motion artifacts and achieve minimal distortion to the denoised ECG signal. In addition, when processing pathological ECG signals, our proposed method should have better performance in preserving pathological characteristics of ECG waveform because it can suppress motion artifacts in the frequency range of 0 to 5 Hz while retaining all frequency components of the pathological ECG waveform. However, this could not be proved in experiments because ECG measurement on wearers who are suffering from heart problem may put wearers at risk, which is contrary to experimental ethics and in any case out of the scope of this research. 

This work provides a solution to the noise problem in wearable ECG measurements. Since the presence of motion artifact in the ECG signal may corrupt the ECG waveform resulting in an incorrect diagnosis of heart disease in medical, or performance in sports and well-being, the motion artifact reduction system proposed in this paper can effectively improve the quality of the ECG signal and reduce the possibility of misdiagnosis. In the recent COVID-19 outbreaks, there was a huge demand for wearable health systems that measure heart rate and respiration [41], and these systems also faced motion artifact interference when measuring vital signs. Our method offers a possible solution to these systems, paving the way for truly wearable ECG systems of wider end uses.

## Figures and Tables

**Figure 1 sensors-22-05493-f001:**
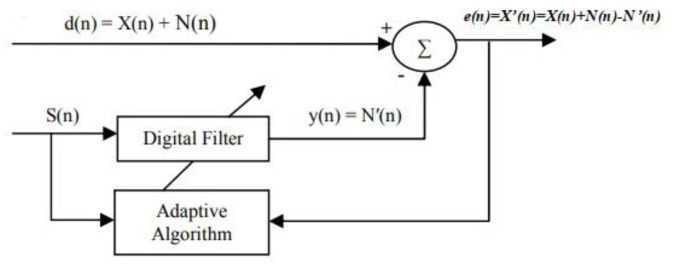
Principle of the adaptive filter in noise cancellation.

**Figure 2 sensors-22-05493-f002:**
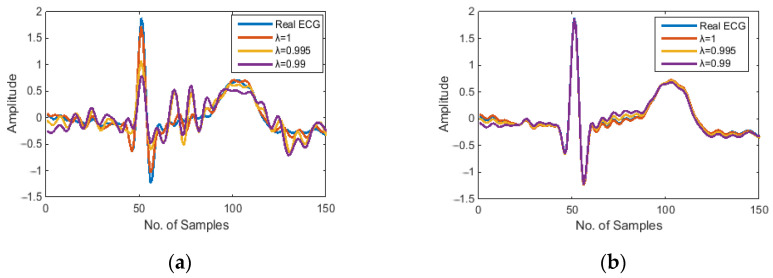
Samples of denoised ECG waveform: (**a**) denoised by RLS adaptive filter; (**b**) denoised by RLS adaptive filter with a 5 Hz low-pass filter.

**Figure 3 sensors-22-05493-f003:**
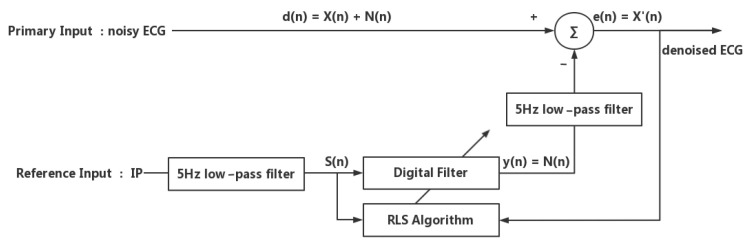
The proposed adaptive motion artifact reduction method.

**Figure 4 sensors-22-05493-f004:**
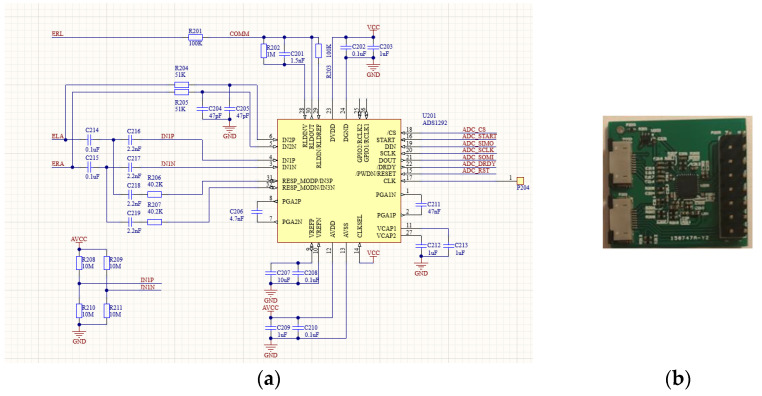
The analog front-end (AFE) circuit board: (**a**) schematic diagram; (**b**) circuit board.

**Figure 5 sensors-22-05493-f005:**
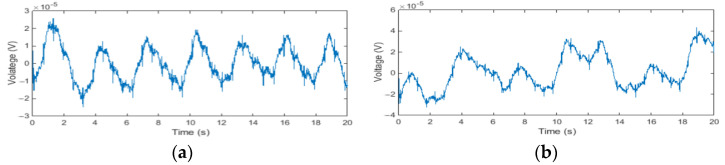
The IP signal recorded by the ECG -IP acquisition system: (**a**) the IP signal contains cardiac-related artifacts; (**b**) the IP signal contains cardiac-related artifacts and motion artifacts.

**Figure 6 sensors-22-05493-f006:**
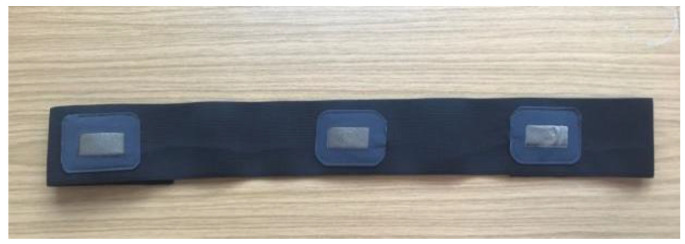
The chest band for wearable ECG measurement.

**Figure 7 sensors-22-05493-f007:**
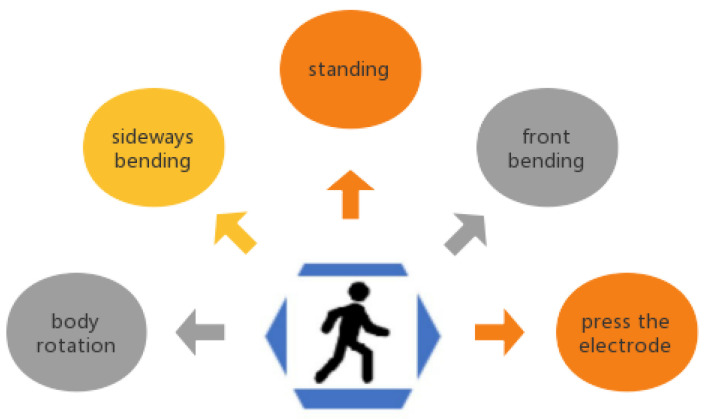
Typical motion types to trigger the motion artifacts.

**Figure 8 sensors-22-05493-f008:**
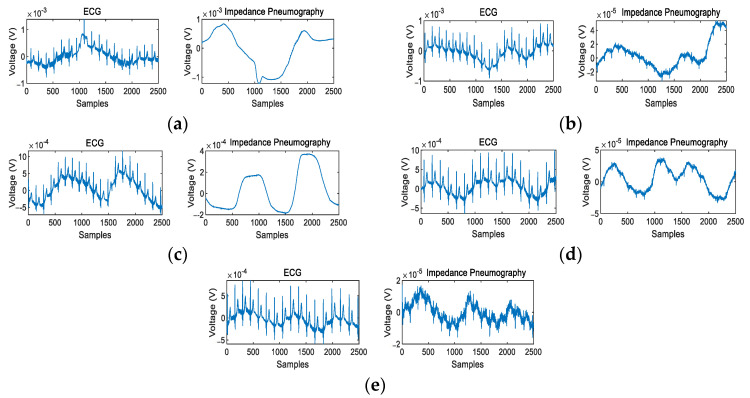
The typical ECG and IP signals recorded under different body motions: (**a**) motion type I, (**b**) motion type II, (**c**) motion type III, (**d**) motion type IV, (**e**) motion type V.

**Figure 9 sensors-22-05493-f009:**
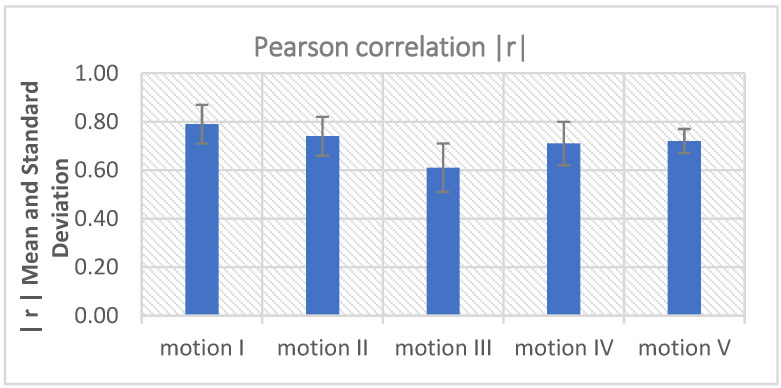
The correlation between low-pass filtered ECG and IP signal.

**Figure 10 sensors-22-05493-f010:**
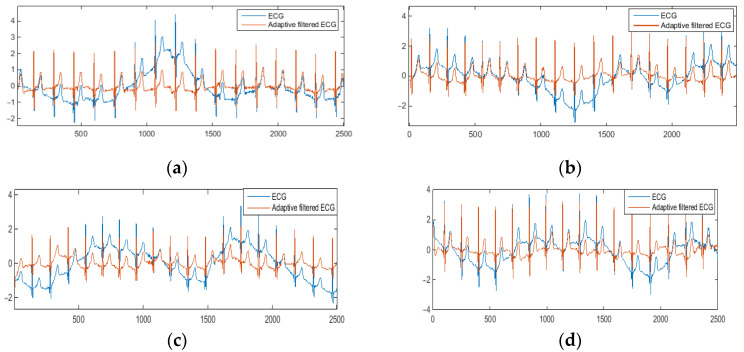
The denoised ECG signal: (**a**) motion type I, (**b**) motion type II, (**c**) motion type III, (**d**) motion type IV, (**e**) motion type V.

**Figure 11 sensors-22-05493-f011:**
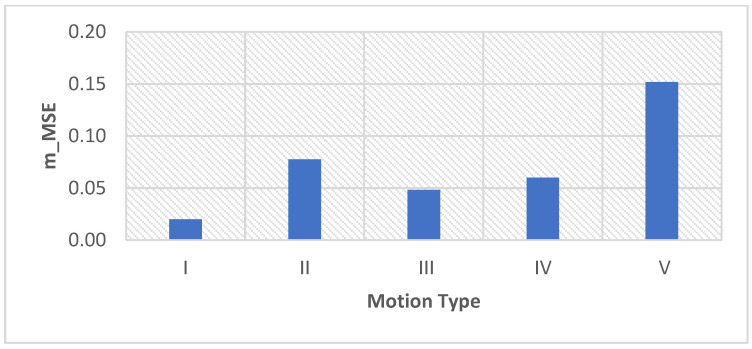
The m_MSE of the denoised ECG signal processed by the proposed method.

**Figure 12 sensors-22-05493-f012:**
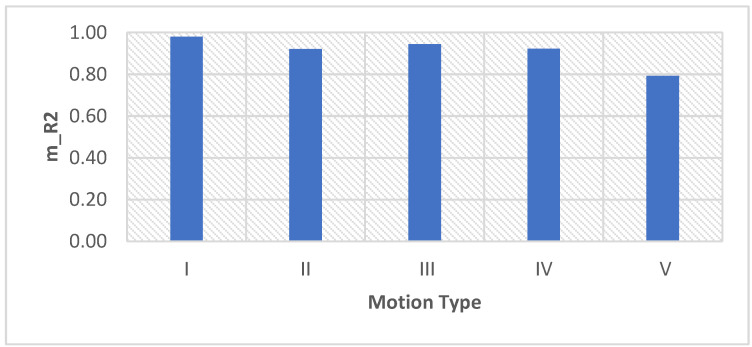
The m_R^2^ of the denoised ECG signal processed by the proposed method.

**Figure 13 sensors-22-05493-f013:**
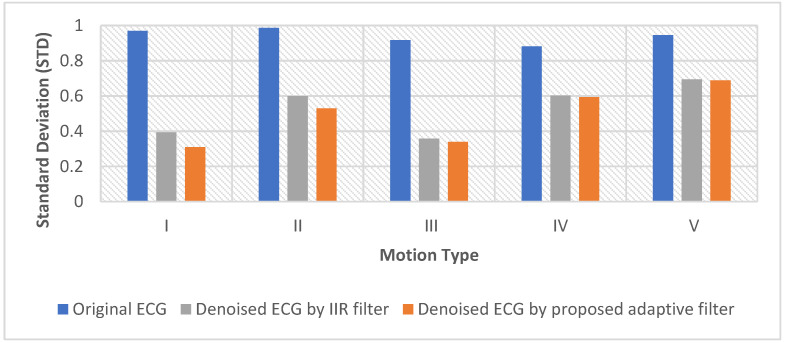
The performance of the IIR filter and the proposed method.

**Table 1 sensors-22-05493-t001:** Different motion artifacts.

Motion Types		Description
I	Pressing of the electrode	One electrode is pressed and released immediately by hand.
II	Upper body anterior flexion and extension (upper body front bending)	Bend the spine forward to 45 degrees and then return upright. Bending speed: about 0.2 cycles/sec
III	Upper body lateral flexion and extension (upper body sideways bending)	Laterally bend the spine to 45 degrees then return to upright. Bending speed: about 0.2 cycles/sec
IV	Upper body rotation	Twist the torso to 90 degrees and then return. Twisting speed: about 0.2 cycles/sec
V	Standing	Stand still and breath normally Respiration rate: about 0.3 Hz

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
