# Peer review of "Adaptive Motion Artifact Reduction in Wearable ECG Measurements Using Impedance Pneumography Signal"

_sensors, 2022, doi:10.3390/s22155493_

Round 1
Reviewer 1 Report
A direct question that comes to mind is what will happen if the 5 Hz low-pass filtered noisy ECG signal is used in the adaptive filtering instead of the the 5 Hz low-pass filtered IP signal. If not possible to perform such an experiment, then a paragraph with a theoretical discussion on the issue is recommended to be added.
A description of the electrodes and how they are placed is missing. Please clarify if the wearable measures 1-lead, 3-leads or 12-leads ECG and if the adaptive filtering performs differently on different leads correlated also to the motioon artifact type.
A discussion on pathologic ECGs is recommended to be added; pathologic ECGs may have different frequency profiles and in some cases useful components around 5Hz. How will the adaptive filtering perform then?
Finally, wearables have been recently used in covid-19 monitoring use cases, where heart rate is also measured. See for example: "IoT-based smart triage of Covid-19 suspicious cases in the Emergency Department" Barbara Fyntanidou, et al., 2020/12/7, 2020 IEEE Globecom Workshops
A discussion on the usefuleness of the adaptive filtering on such cases is recommended to be added, along with a reference to the paper above.
Reviewer 2 Report
Figure 1 corresponds to the article “Comparison of Motion Artefact Reduction Methods and the Implementation of Adaptive Motion Artefact Reduction in Wearable Electrocardiogram Monitoring”. Since the article corresponds to the two authors Xiang An and George K. Stylios, it should be cited in Figure 1. Also, the overall results are based on the article “Comparison of Motion Artefact Reduction Methods and the Implementation of Adaptive Motion Artefact Reduction in Wearable Electrocardiogram Monitoring”, so this should have been specified.
For Figure 1 it should also be pointed out that for example, when y(n) equals or approaches the noise N'(n) ≈ N(n) in the corrupted signal, i.e. y(n) ≈ N'(n), the error signal e(n) will approach the clean signal X'(n). Therefore, the noise is cancelled. It should also be specified that e(n) is equal to d(n)-y(n) and the adaptive algorithm will update the coefficients of the filter until y(n) equal or nearly equal to: N'(n) ≈ N(n).
Figure 3 is unclear.
Eg. 1 it could have been written that the Pearson correlation coefficient is more accurately described by the formula:
Where, cov(x,y) is the covariance and E is is the expectation.
Reviewer 3 Report
There are some obvious referencing errors in the paper, for example "Floyd [4] found that" should be "Floyd and Keele [4] found that"; "Raya et al. [26] attached" should be "Raya and Sison [26] attached", this issue is throughout the paper.
Figure 1 is incorrectly placed before it is discussed in the main text.
Figure 3 seems a poor-quality bitmap that I cannot read.
Figure 6 is readable, but again a poor-quality bitmap.
Table 1 is presented before it is discussed in the main text.
The paper states "d lowpass(n)" after equation 2 and 3, yet there is no (n) in those equations.
Figure 9 is presented before it is discussed in the main text.
The reference list makes no sense at all. For example, "Biomedical Engineering, IEEE Transactions on"? Why is this not represented as its correct title of "IEEE Transactions on Biomedical Engineering"? There are many other issues with the reference list.
After reading the paper, despite the really obvious editorial issues, the authors do have some interesting ideas. However, I am not convinced of the novelty of the idea. Addressing motion artifacts in ECG has been a long-standing issue and there are many research papers on the topic. This paper does not seem to present a good understanding of the current state of the art, therefore novelty is just not convincing.
I purpose the authors address the editorial issues, and significantly improve their literature coverage in order to then demonstrate novelty. However, I do look forward to their resubmission in due course.
Round 2
Reviewer 3 Report
It seems the authors addressed my past concerns.